# Flooding in the Digital Twin Earth: The Case Study of the Enza River Levee Breach in December 2017



Angelica Tarpanelli [1,*], Bianca Bonaccorsi [1,2], Marco Sinagra [3], Alessio Domeneghetti [4], Luca Brocca [1] and Silvia Barbetta [1]

1    Research Institute for Geo-Hydrological Protection, National Research Council, Via Madonna Alta 126, 06128 Perugia, Italy; bianca.bonaccorsi@irpi.cnr.it (B.B.); luca.brocca@irpi.cnr.it (L.B.); silvia.barbetta@irpi.cnr.it (S.B.)
2    Department of Engineering, University of Messina, Contrada di Dio, Sant'Agata, 98158 Messina, Italy
3    Department of Engineering, University of Palermo, Viale delle Scienze, 90128 Palermo, Italy; marco.sinagra@unipa.it
4    Department of Civil, Chemical, Environmental and Materials Engineering, University of Bologna, Viale del Risorgimento, 2, 40136 Bologna, Italy; alessio.domeneghetti@unibo.it
*    Correspondence: angelica.tarpanelli@irpi.cnr.it; Tel.: +39-075-501-4426

**Abstract:** The accurate delineation of flood hazard maps is a key element of flood risk management policy. Flood inundation models are fundamental for reproducing the boundaries of flood-prone areas, but their calibration is limited to the information available on the areas affected by inundation during observed flood events (typically fragmentary photo, video or partial surveys). In recent years, Earth Observation data have supported flood monitoring and emergency response (e.g., the Copernicus Emergency Service) thanks to the proliferation of available satellite sensors, also at high spatial resolution. Under this umbrella, the study investigates a levee breach that occurred in December 2017 along the Enza River, a right tributary of the Po River, that caused the inundation of a large area including Lentigione village. The flood event is simulated with a 2D hydraulic model using satellite images to calibrate the roughness coefficients. The results show that the processing and the timing of the high-resolution satellite imagery is fundamental for a reliable representation of the flooded area.

**Keywords:** rivers; inundation; remote sensing; Digital Twin Earth; Sentinel-1; Sentinel-2; levee break





## 1. Introduction

Flood risk mitigation policies require accurate delineation of flood hazard maps. In the literature, several methods have been implemented. They are based on empirical models [1] or on numerical hydrological and hydrodynamic models [2,3]. This second approach to define flood-prone areas is characterised by the use of rainfall-runoff transformation to estimate river discharge and flow propagation in the area using one-dimensional (1D) or two-dimensional (2D) hydraulic models. Typically, the use of the hydrological/hydraulic models is associated with parameter calibration [4], which requires historical hydrometric measurements or assimilation techniques to obtain good performance. In the absence of recorded data, fragmentary ground/remote data (e.g., pictures, videos, direct testimony, information derived from helicopter video, etc.) are used as a benchmark [5]. In recent decades, there have been major advances in remote sensing for flood hazard delineation [6–9]. For example, satellite remote sensing technology is used to support the management and monitoring of flood events [10,11] because it provides repeated information at different spatial and temporal resolutions on the location, extent and dynamic evolution of flow. In particular, Synthetic Aperture Radar (SAR) data are often employed to estimate water levels and flooded areas [12,13]. Optical band satellite imagery also provide reliable and easy-to-interpret results [9], but it is rather difficult to obtain images during

a flood event due to cloud cover [14]. For an updated review of the studies using Earth Observation (EO) data for flood modelling and prediction, the reader is referred to [15].

Based on the large trustworthiness of satellite observations, the European Copernicus developed the Emergency Management Service [16], a rapid mapping service for the production of damage assessment maps caused by natural or man-made disasters, to support the civil protection services of the Member States. The Copernicus Emergency Management Service has two components: (1) mapping and (2) early warning; it is free of charge to all users both in urgent mode, for emergency management activities, and in standard mode, to support environmental disaster management activities. This is carried out by analyzing pre-disaster risk assessment and vulnerability of population or post-disaster recovery and reconstruction. Mapping is implemented by the European Commission's Directorate-General (DG) Joint Research Centre (JRC) through a service that has been operational on a global scale since 1 April 2012. The products generated by the service are maps based on satellite imagery and provided in digital and/or vector formats, to be combined with other data sources (e.g., digital feature sets in a geographic information system) for geospatial analysis to support decision-making by emergency managers.

Concurrently, the European Commission aims to develop a highly accurate digital model of the Earth on a global scale to monitor, simulate and predict the interaction between natural phenomena and human activities under the Destination Earth (DestinE) initiative [17]. Space agencies propose activities on the use of their satellites to generate products that contribute to the development of the Digital Twin Earth (DTE). A first example is the DTE for Hydrology, which integrates high-resolution satellite observations with advanced hydrological and hydraulic modelling—including anthropogenic factors—to develop a 4D reconstruction of the water cycle [18,19].

On this basis, the present work is part of the activities of DTE-Hydrology, investigating the use of open-access high-resolution data (Sentinel constellations) in combination with a flood inundation model to replicate a flood event that occurred in the Po River basin due to a levee failure. Unlike previous studies in the literature [20–22], the present work focuses more on the prospect of using modelling and the satellite images to build a complete digital product. To this end, we will assess the potential and limitations of such a system with a look at the still missing points that would allow its operational use by stakeholders.

In the following, Section 2 describes the ground and satellite data used to build the case study, the model used in the analysis, and the calibration procedure; Section 3 presents the results obtained and Section 4 presents a discussion highlighting the importance of the timeliness of satellite imagery and the role of the Copernicus Emergency maps available for the same area.

## 2. Materials and Methods

This section contains a brief description of the material collected by ground networks during the levee breach, the satellite data retrieved from optical and SAR sensors, the 2D hydraulic model used and the procedure for the calibration of the model by the use of satellite images.

### 2.1. The Levee Failure Event and the Collected In Situ Observations

The study area is located in the Enza River Basin, a right tributary of the Po River in Northern Italy, with an area of 890 km$^2$ and a river length of about 100 km (Figure 1). The Enza River is a torrential river with flood events in autumn and spring and low flow in winter, while it is typically almost dry in summer. The morphological and lithological characteristics of the basin, such as the shape and the average slope steepness, imply short runoff times, with rapid flood formation and high discharge values. The morphological characteristics of the basin show that most of the tributaries are located between the altitudes of 600 and 250 m a.s.l.; consequently, the flows that cause the highest hydrometric conditions for the stretch of interest are those that correlate with rainfall maxima concentrated in the central part of the basin. The Sorbolo station is the ground hydrometric site

along the Enza River closest to the location of the levee breach. Specifically, the gauged site, where the water level hydrograph is recorded, is located about 5 km upstream of the levee failure and the Lentigione urbanized area. The river discharges calculated by the available rating curve during the flood event occurred in December 2017 are applied as input conditions to the hydraulic modelling (Figure 1).

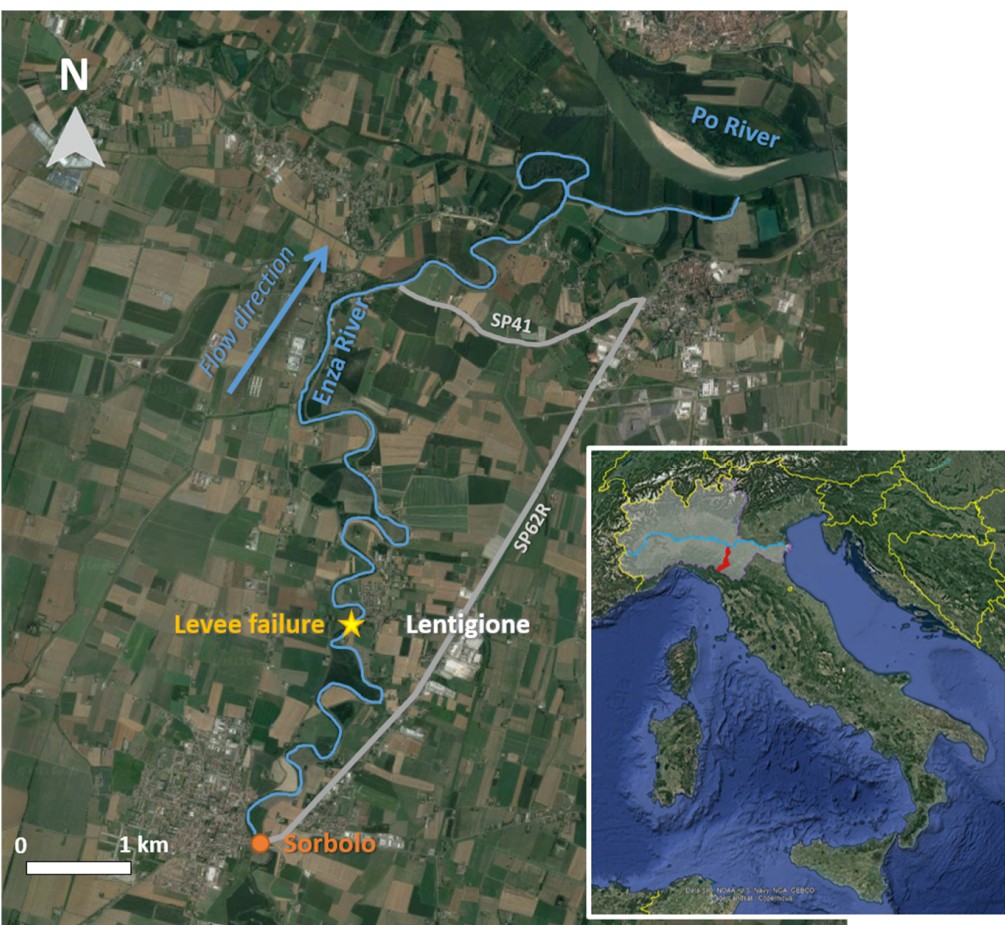

**Figure 1.** Study area of Enza River from the gauged station of Sorbolo to the confluence with the Po River. The village of Lentigione and the location of the levee failure are illustrated.

From 8 to 12 December 2017, an exceptional meteorological event hit the study region, causing significant damage in different areas of the Emilia-Romagna region. The rainfall event was characterised by two distinct pulses of rainfall. At the beginning, for a period of about 48 h, the rainfall was regular and very heavy. This first event saturated the soils and caused heavy runoff in the watercourses, which became critical with the second event. Indeed, the gradual increase in high-altitude temperatures between 11 and 12 December, caused by south-westerly flow, also led to a partial melting of the snowpack in the central-western basins, which further contributed to the total inflow to the rivers. For the second event that occurred between 10 and 12 December, the rainfall was very intense with values greater than 80 mm in 6 h and reaching cumulative values greater than 300 mm in 40 h. At Sorbolo, the recorded water level reached the historical maximum (available since 1935) of 12.17 m at 7:30 a.m. on 12 December. The flood event with a duration of 12 h was estimated with a return period of 200 years (Figure 2).

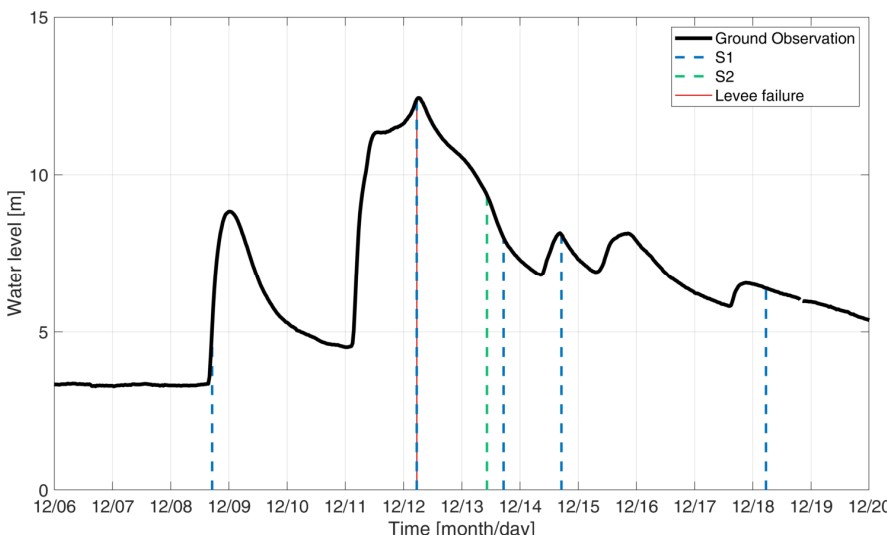

**Figure 2.** Discharge hydrograph recorded at Sorbolo gauged section. The occurrence time of the levee breach and the satellite images (S1, Sentinel-1; S2, Sentinel-2) available during the event.

On the morning of 12 December, at around 5.30 a.m., the right bank failed near the village of Lentigione di Brescello (Reggio Emilia). The initial overtopping occurred at three adjacent points along the embankment, initially triggering three very close breaches, which almost merged into a single large one in time. The overtopped part of the levee was 250 m long and the final breach width was approximately 160 m [23]. A large amount of water was released in the surrounding area through the breach. At 2 p.m., the flooded volume was estimated to be around 10 million m$^3$ and the subsequent propagation of the flood wave from the Enza was expected to release a volume of the same order over the next 36 h. The flooding affected 470 buildings, of which 449 were residential, 5 were places of worship and 16 were businesses [24].

*2.2. Satellite Images*

For the analysis, we selected satellite images from Sentinel-1 and Sentinel-2, which to date are the satellites that provide the highest resolution images at no cost to the operator. The two satellites provide SAR and optical images, respectively, which are recognized in the scientific literature as tools for analysing the extent of flooding. During the selected flood event, several Sentinel-1 and 2 images were available as shown in Figure 2, where the acquisition time of each satellite image is represented by dashed lines. It is obvious that the cloud cover during the rain event prevented the acquisition of the optical Sentinel-2 image, while several Sentinel-1 scenes were available to observe the evolution of the flooding. Although six images observed the flood event, only three of them were analysed and only two (one optical and one SAR) were used to calibrate the roughness parameter of the hydraulic modelling. In fact, the flood was triggered at the peak value recorded on 12 December and is partially visible in the SAR image corresponding to the peak, while it is clearly visible in the two images of 13 December, as will be seen below. The satellite missions and the image processing used to derive the flooded area are described below.

2.2.1. Sentinel-1 Images and Processing

Sentinel-1 is a C-band Synthetic Aperture Radar (SAR) consisting of two polar-orbiting satellites. As an active sensor, SAR is able to penetrate clouds and provide images in all weather conditions, day and night. The sensors have a temporal resolution of 6 days in constellation mode (12 days for single mission) and a spatial resolution of 10 m. In SAR images, smooth open water is represented by law backscatter values with pixels appearing in dark tonality because all the radar radiation is reflected away from the sensor. However,

distinguishing flooded areas remains difficult, especially when the water surface becomes rough due to the presence of vegetation or windy conditions [15].

Two Sentinel-1 images were selected to study the flood event: (1) Sentinel-1A GRD in VV polarization, descending, acquired on 12 December 2017 at 5:27 a.m. (Figure 3a); (2) Sentinel-1A GRD in VV polarization, ascending, acquired on 13 December 2017 at 5:15 p.m. (Figure 3b).

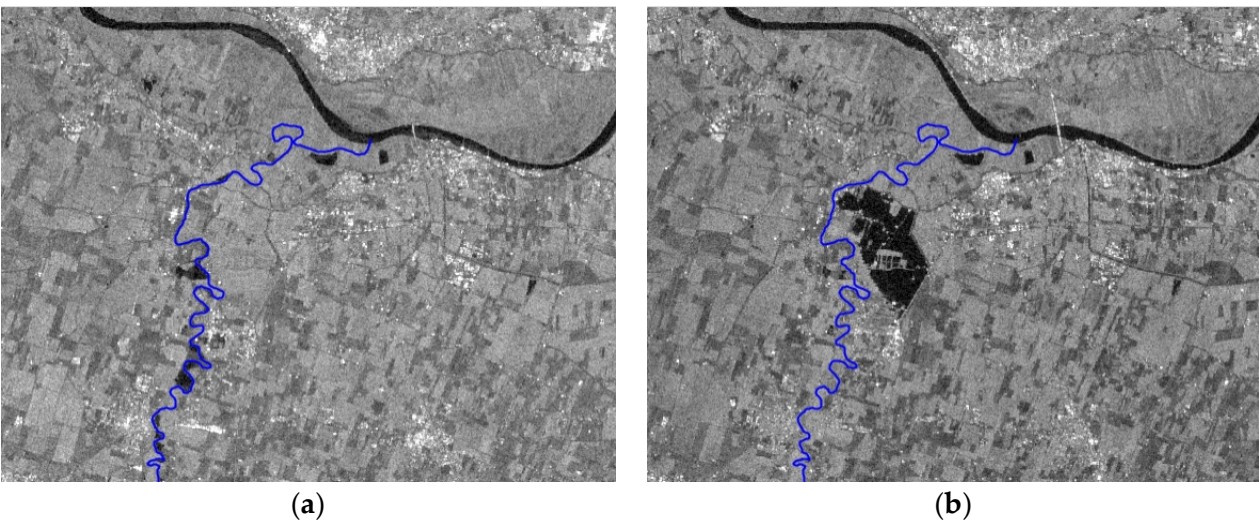

(**a**)  (**b**)

**Figure 3.** Satellite SAR images from Sentinel-1A: (**a**) GRD in VV polarization, descending on 12th December 2017 at 5:27 a.m.; (**b**) GRD in VV polarization, ascending on 13th December 2017 at 5:15 p.m. In blue, the shapefile of the river is also illustrated.

For the SAR images, the histogram threshold [25] is a consolidated approach used to discriminate between flooded and non-flooded areas, using the radiometric distributions of water bodies and other land-use types.

### 2.2.2. Sentinel-2 Image and Processing

The sensors on board Sentinel-2 cover 13 spectral bands with spatial resolutions of 10, 20 and 60 m in the visible, near-infrared and shortwave-infrared. Sentinel-2 imagery can be severely affected by cloud cover and can only be used during daylight hours and in good weather conditions. These conditions make the number of opportunities to detect the maximum extent of a flood event rather low (27% against 55% of the SAR mission in case of a small basin and very-high flood event according to [14]). However, the levee failure was a rather significant event, affecting a large part of the floodplain around the river and the water remained in the inundated areas for several days. This combination of factors allowed us to collect an optical image: Sentinel-2A, descending, acquired on at 10:24 a.m. on 13 December 2017. This image shows the state of the area about 29 h after the levee breach occurrence (Figure 4).

The extraction of the flooded areas from the Sentinel-2 optical satellite image was carried out according to the well-established methods available in the literature and summarized in the study by [26]. They were the Automated Water Extraction Index (AWEI, [27]) in the two versions "nsh", where dark built surfaces in urban areas are removed and "sh", where also shadow pixels are removed; Normalized Difference Water Index (NDWI, [28]), its modified version (mNDWI, [29]), the Normalized Difference Flooding Index (NDFI, [30]) and the Water Ratio Index (WRI, [31]). These spectral indices were calculated based on the combination of the bands BLUE (490 nm), GREEN (560 nm), RED (665 nm), NIR (842 nm), SWIR1 (1610 nm) and SWIR2 (2190 nm) as follows:

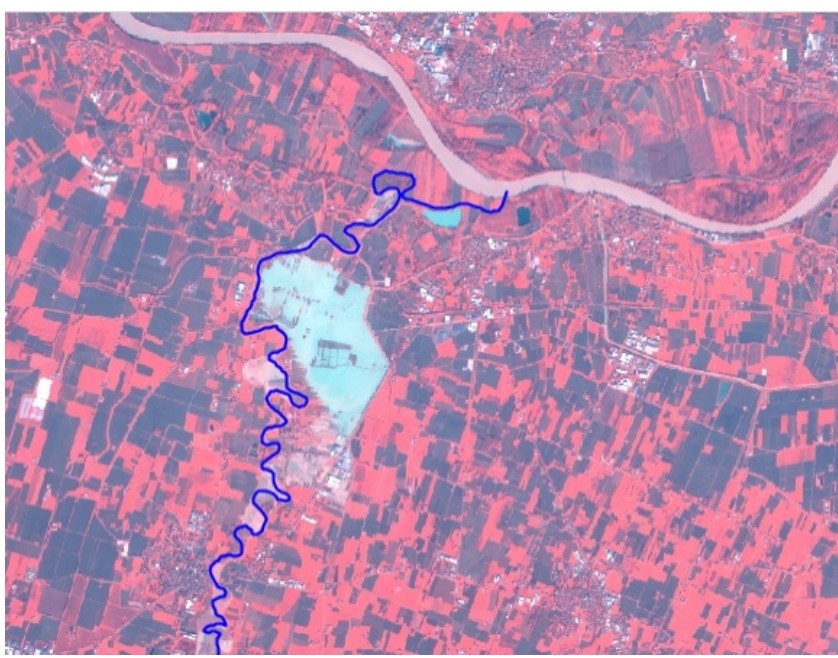

**Figure 4.** Sentinel-2A, descending, acquired on 13th December 2017 at 10:24 a.m. The representation is R: NIR; G: Green; B: Blu. In blue, the shapefile of the river is also illustrated.

$$AWEI_{nsh} = 4 \times (GREEN - SWIR1) - (0.25 \times NIR + 2.75 \times SWIR2) \tag{1}$$

$$AWEI_{sh} = BLUE + 2.5 \times GREEN - 1.5 \times (NIR + SWIR1) - 0.25 \times SWIR2 \tag{2}$$

$$NDWI = (GREEN - NIR)/(GREEN + NIR) \tag{3}$$

$$mNDWI = (GREEN - SWIR1)/(GREEN + SWIR1) \tag{4}$$

$$NDFI = (RED - SWIR2)/(RED + SWIR2) \tag{5}$$

$$WRI = (GREEN + RED)/(NIR + SWIR1) \tag{6}$$

Based on these spectral indices and assuming a threshold equal to 0 for all indices, except for NDFI (which is equal to 0.32) and WRI (which is equal to 1), the flooded areas were derived and used successively to calibrate the model parameters, as explained in Section 2.4.

*2.3. Flood Model*

2.3.1. The 2D Hydraulic Model

The 2D hydraulic model employed for flooding analysis is named WEC-Flood and is suitable for flood hazard mapping in areas characterised by high urbanization and complex topography [32]. The model adopts the diffusive wave approximation of the Saint Venant Equations to obtain a system of differential equations in the 1D and 2D computational domains, which has several advantages compared to dynamic modelling. Among these: (1) with the same simulated time and computational capacity, it guarantees the solution in much faster times than the complete modelling; (2) the diffusive model in the 2D domain is characterised by a smaller sensitivity of the computed water height with respect to the topographical error. To solve the Equations, an unstructured hybrid mesh is used; the mesh is identified by the Hydronet algorithm [32], which uses as input the DEM data of a

rectangular area and other topographical information, such as the river bed and the levee crest elevations.

1D channels are discretised using quadrilateral elements, with one couple of opposite edges overlapping the trace of two river sections and the other couple connecting their ends. The trace of each river section is extended up to the minimum topographic elevation, where 1D flow conditions are expected. The 2D computational domain is given by the whole area of the catchment and is discretised by triangular elements satisfying the Generalized Delaunay conditions [33]. The model is solved in the context of the MAST (MArching in Space and Time) approach [34], where the solution at the end of each time step is sought after through a fractional time-step procedure, splitting the original problem in a prediction plus a correction sub-problem.

### 2.3.2. Set up of the Model

The model requires as input data: (1) a detailed DEM of the area to be investigated; (2) additional topographic information representing hydraulic singularities as culverts, levees and bridges, to consider their impact on the flow dynamics; (3) upstream and downstream boundary conditions, i.e., discharge hydrographs observed at gauged sections or estimated by hydrological models, water levels at the confluence, rating curve relationships, etc. The expected outputs are (1) flooded maps identifying the area affected by flooding, (2) the water depth value at the different nodes of the computational grid and (3) the discharge hydrograph at different sections selected along the main channel, including the outlet one.

The first step in the implementation of the WEC-Flood model is the collection of shapefiles and raster data, which are essential to define the computational mesh. In particular, two shapefiles define the riverbed and the levee locations; while the raster is the 1 m resolution Digital Elevation Model (DEM), provided by the Emilia Romagna Region. The original DEM is modified considering the riverbed and embankment elevations derived from fourteen topographic sections surveyed along the simulated reach of the Enza River, which is about 13 km long.

The modified DEM and the shapefiles are successively used as input to identify a mesh with a spatial resolution of 5 m, in which the computational nodes are forced to be located along the river bed and the levees. This upscaling does not lead to any deterioration in the analysis, since the heights of the main channel and the banks have been maintained and, for the floodplain, the comparison made later with the satellite images refers to a 20 m resolution. The upstream boundary condition is represented by the discharge hydrograph recorded at Sorbolo gauged station (Figure 2), provided by the Dexter system of the Arpae Emilia Romagna [35], while the downstream boundary condition is hypothesised as a null diffusive condition.

Moreover, the WEC-Flood model is quite flexible and allows for the simulation of a rapid change in the topography as the levee failure occurred during the selected flood event. To this end, the topographical elevation of selected nodes of the mesh developed by Hydronet is modified at a lower value corresponding to the lowest elevation of the breach, at the time of the failure occurrence (5:30 a.m.), with a geometry of 160 m long and 3 m deep, equivalent to the three breaches effectively generated.

### 2.4. Calibration of the Hydraulic Model with the Satellite Images

The use of a 2D model to simulate flood events allows one to consider the complex topographic effects of the floodplain and calibrate in detail the domain parameters, mainly the roughness. In general, the calibration of such a parameter is carried out by reproducing the hydrograph of the water level or the river discharge in the downstream section. In this case study, no gauging station is available at the downstream section (only Sorbolo at the upstream section), and the information on the flooded areas is derived from fragmentary ground/remote data (e.g., pictures, videos, direct testimonies, indications derived from videos recorded during helicopter flights, etc.). The use of high-resolution Sentinel satellite

imagery represents a significant improvement for the 2D model calibration. In this context, the calibration of the roughness parameter $n$ (i.e., the Manning coefficient) is carried out by comparing the simulated flooded areas with those observed by satellite. To cover a range of plausible values, the roughness coefficient is varied in a flexible range between 0.06 and 0.13 m$^{-1/3}$s, with a step of 0.01 m$^{-1/3}$s for a total of 8 simulations, assuming a uniform coefficient for all computational domains. It is worth noting that preliminary simulations were carried out to test the effects of using different values of the $n$ parameter for the main channel elements and the floodplains and embankment elements; the results clearly show that no significant changes are introduced by differentiating the values.

The comparison between each simulated flood scenario and the satellite-derived maps at the same time step was performed using the procedure proposed by Aronica et al. [36], based on the index $F$, which represents the measure of the overlap between the observed and computed flooded areas, as follows:

$$F = \frac{A}{A + B + C} \tag{7}$$

where $A$ is the amount of wet area correctly predicted by the hydraulic model, $B$ is the area predicted to be wet that is instead observed dry (overprediction) and $C$ is the wet area not predicted by the model (underprediction). $F$ ranges from 0 to 1; if it is equal to 1, observed and predicted areas exactly match, and if it is equal to 0, there is no overlap between the predicted and observed areas. Therefore, maximising $F$ allows one to estimate the optimal value of $n$ for which the simulated flooded area is as close as possible to the observed one [37].

## 3. Results

This section shows the results obtained by modelling the levee failure and the consequent floodplain inundation.

### 3.1. Satellite-Derived Flooded Area Maps

In the case of the two SAR images acquired on 12 December 2017 and on 13 December 2017, the flooded areas cover different areas and the histogram is quite different, as shown in Figure 5a,b. However, a common threshold can be identified by looking at the local minimum value between the two distributions, and it is set to $-18.1$ dB. The resulting flooded areas, with values below the threshold, are shown in Figure 5c,d and represent the best compromise between the two images. Indeed, the first available image (Figure 5c) identifies the initial ponding areas close to the meanders of the river and not close to the levee breach, confirming that the failure occurred after 5:27 a.m. (estimated at 5:30 a.m.). The second image acquired after the peak of the river discharge (Figure 5d) does not show the inundation along the river, but only the large volume of water stored between the SP41 and SP 62R roads (Figure 1). Because the first image does not reproduce an effective flooded area, for the calibration of the hydraulic parameter, we considered the second SAR image only (Figure 3b).

The analysis of the Sentinel−2 imagery results in six flooded areas as shown in Figure 6. In general, the maps show a very similar delineation of the flooded areas, except for the mNDWI index, which shows a larger flooded area. In fact, when compared with the false colour image in Figure 4, the water map extracted with mNDWI seems to be the most reliable, showing even the wet condition of the areas in the upstream part (in the South) of the investigated river reach. For this reason, this map was used for the calibration of the hydraulic model.

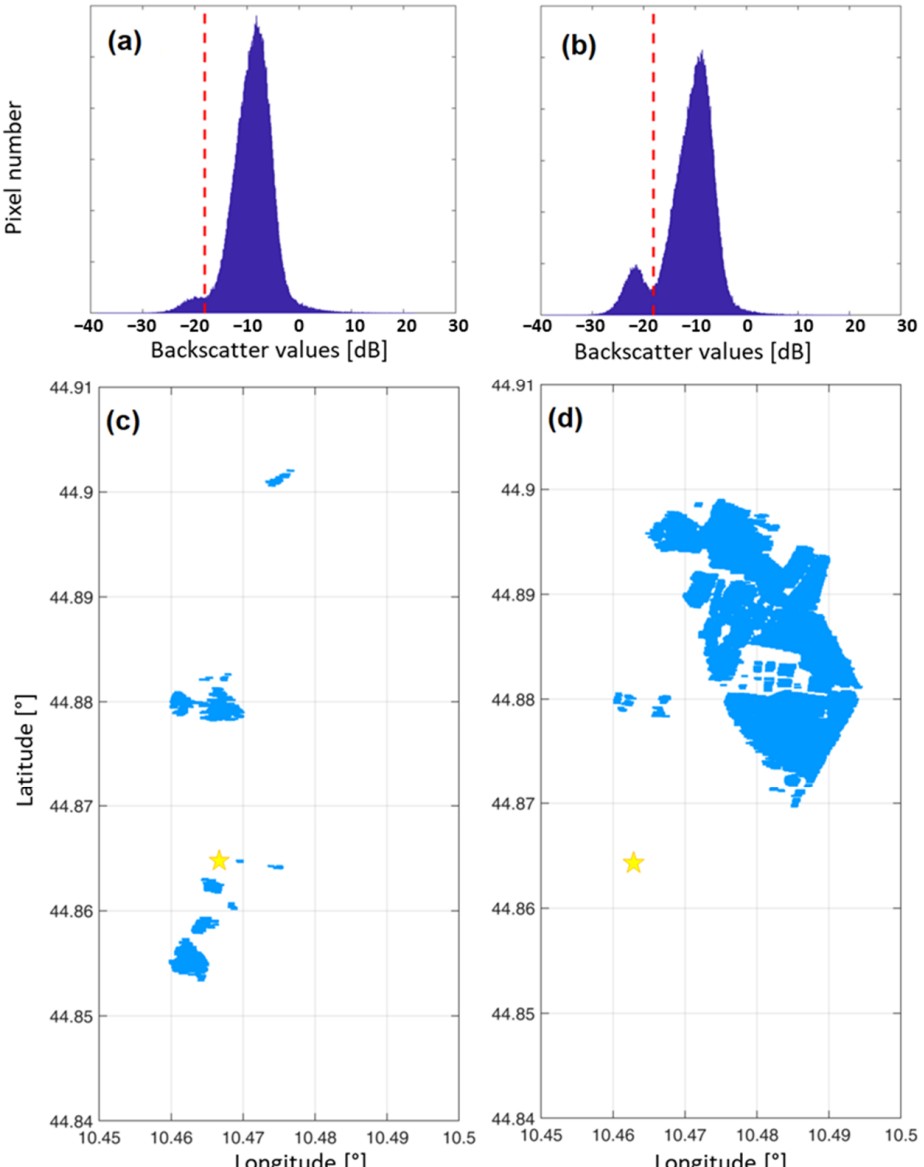

**Figure 5.** Histogram of the backscatter (**a**,**b**) and flooded areas (**c**,**d**) below the threshold shown in dashed red line of the two SAR images acquired on 12 December, 2017 at 5:27 a.m. (**a**,**c**) and 13 December 2017 at 5:15 p.m. (**b**,**d**). Yellow star indicates the location of the levee breach.

### 3.2. Dynamic Flood Modelling Results

The WEC-Flood model was run for each Manning roughness value, for a total of 8 runs. To compare the flood maps with the satellite observations, the simulations were stopped at the same time frame of the satellite acquisition image. Due to the nature of the model, the resulting maps represent the maximum extent of the flooded area simulated up to that specific time frame. This can be accurate during the rising limb of the hydrograph, where the inundation has hit an increasing area, whereas during the receding limb of the hydrograph, the model is unable to simulate the dry phase.

Figure 7 shows the F index (calculated by Equation (7)) derived by comparing the hydraulic model result with the two satellite-derived maps from the optical (a) and SAR (b) images. The two plots show some differences both in the F-value (for the optical image F values are higher than those of the SAR image) and the optimal Manning roughness coefficient. In particular, the hydraulic model is in better agreement with the flooded area extracted from the optical image. Regarding the comparison with the SAR image, the

WEC-Flood model shows flooded areas in the upstream part that are completely absent in the SAR image (Figure 3b), and this provides a lower number of pixels in agreement (and consequently a lower value of A and a higher value of B in the F formulation).

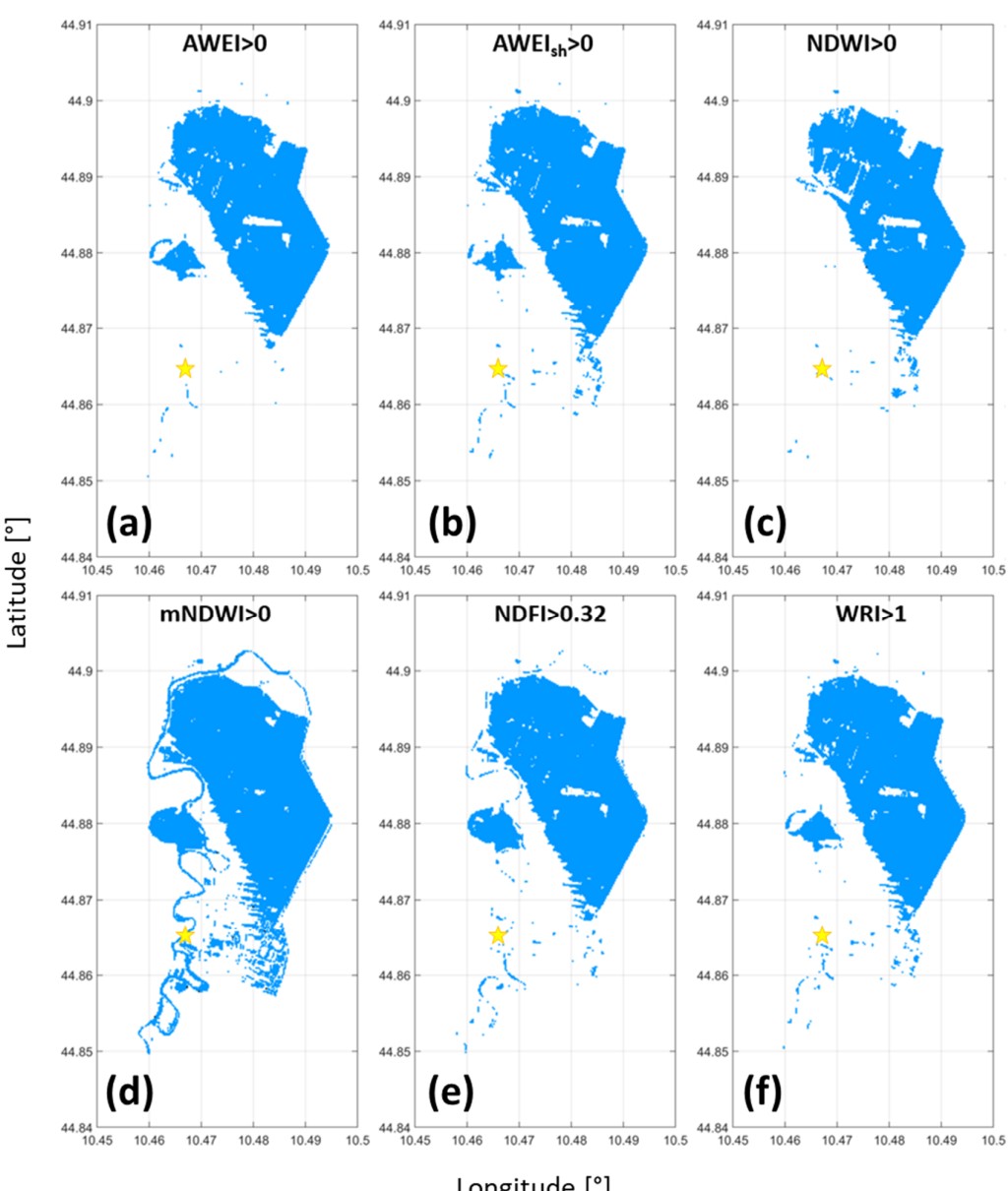

**Figure 6.** Flooded area maps retrieved from the optical image acquired by Sentinel-2 through the use of six different spectral band combinations (see Equations (1)–(6)): (**a**) AWEI_{nsh}, (**b**) AWEI_{sh}, (**c**) NDWI, (**d**) mNDWI, (**e**) NDFI, (**f**) WRI. Yellow star indicates the location of the levee breach.

As a result, the analysis of all simulations indicates that the roughness coefficient equal to 0.11 m$^{-1/3}$s is the one that provides the best reproduction of the flooded area as shown in the Sentinel-2A image, acquired on 13th December 2017 at 10:24 a.m., about 29 h after the formation of the levee breach. Figure 8 shows the comparison between the flooded area simulated by setting $n = 0.11$ m$^{-1/3}$s and the flooded area shown in the optical satellite image.

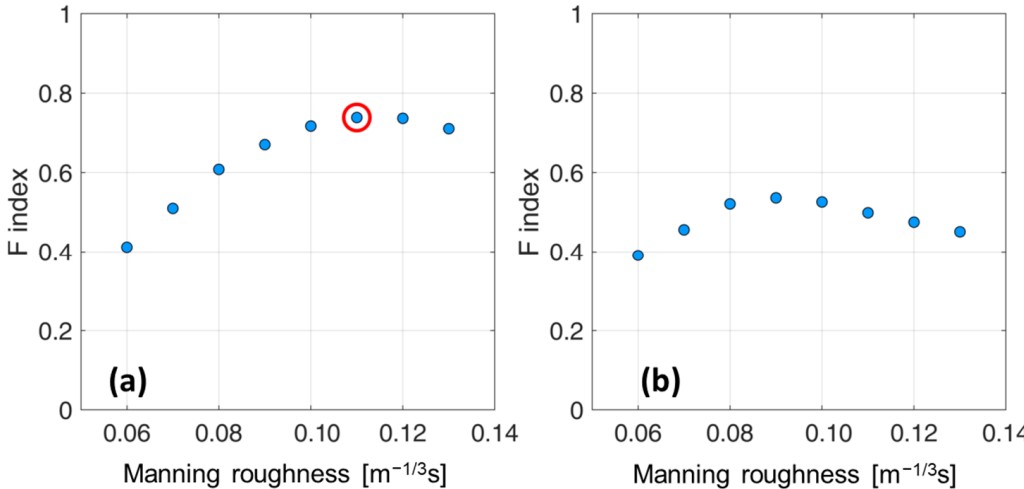

**Figure 7.** Variability of the measure of fit F as a function of the Manning roughness coefficient calculated with respect to the (**a**) satellite optical image acquired on 13th December 2017 at 10:24 a.m. and (**b**) SAR image acquired on 13th December 2017 at 5:15 p.m. Red circle in the subplot (**a**) highlights the optimal manning roughness coefficient selected for the analysis.

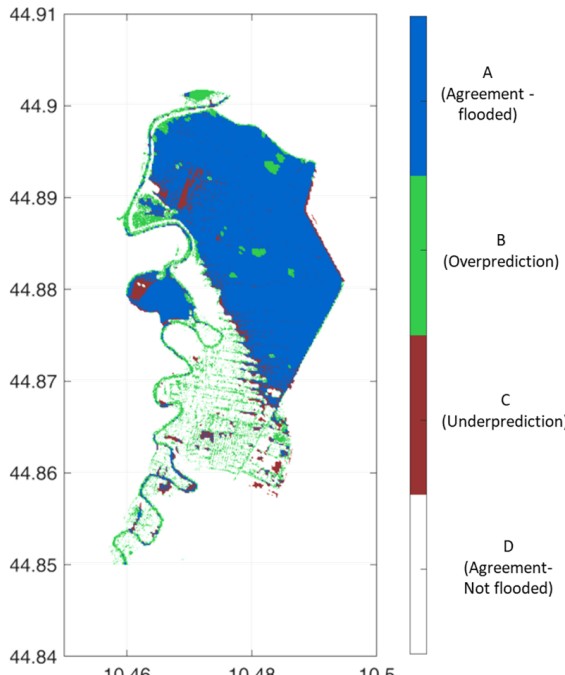

**Figure 8.** 13/12/17 at 10.24 a.m.: comparison between flooded area extracted from the Sentinel 2A satellite image and derived from Wec-Flood model simulation with $n = 0.11$ m$^{-1/3}$s.

By the comparison with the SAR image (acquired by Sentinel 1A on 13th December 2017 at 5:15 p.m., about 36 h after the levee breach occurrence) shown in Figure 9, the roughness coefficient equal to 0.09 m$^{-1/3}$s is the one that provides a better reproduction of the inundated area based on the F index. However, the flooded area simulated by the model, setting $n = 0.09$ m$^{-1/3}$s, largely overestimates the flooded area obtained from the satellite image.

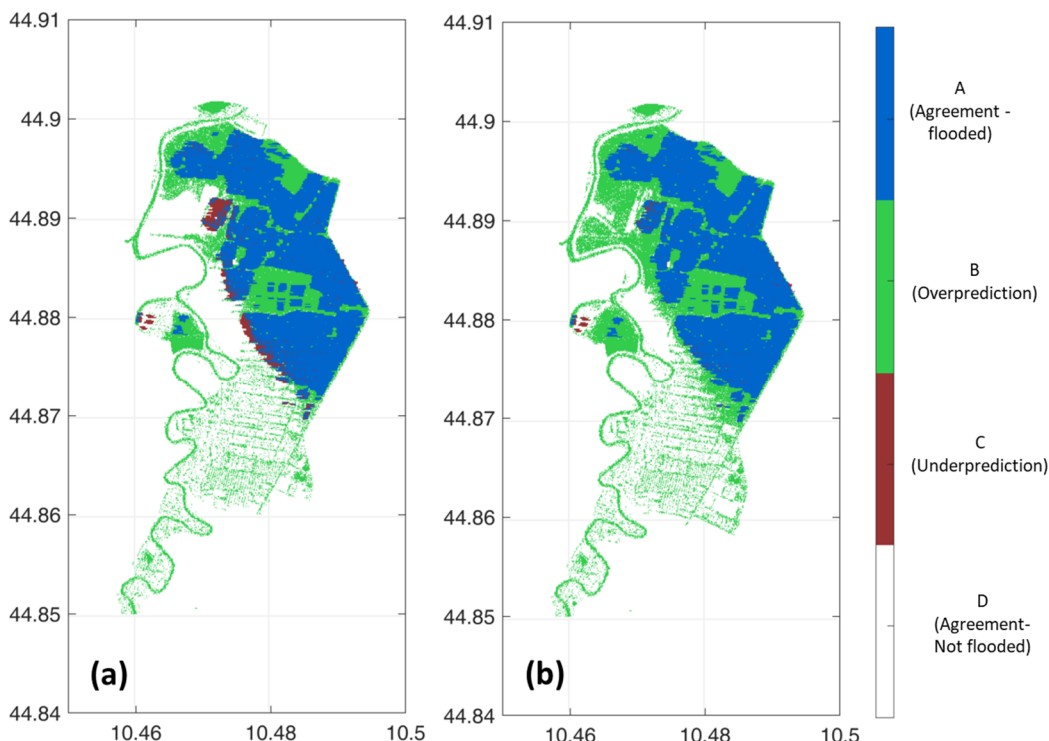

**Figure 9.** 13/12/17 at 5.15 p.m.: comparison between flooded area extracted from the Sentinel 1A satellite image and derived from Wec-Flood model simulation with (**a**) $n = 0.09$ m$^{-1/3}$s and (**b**) $n = 0.11$ m$^{-1/3}$s.

For the sake of comparison, Figure 9 shows also the resulting map with $n = 0.11$ m$^{-1/3}$s optimized from the optical image. The comparison between the two maps shows again a large overestimation of the model: the flooded area extracted from the SAR image is completely included in the results of the model. This result is due to the inability of the model to simulate the drying process that occurred a few hours after the inundation.

*3.3. Comparison with the Copernicus Emergency Service Flood Map*

With regards to the documentation available from the Copernicus Emergency Service for this event, Figure 10a shows a collection of vectors of inundation delineations extracted from the event classified as EMSR260 in Cicognara, Sorbolo and Sant Ilario d'Enza [38]. The final map is accurate and shows the areas affected by the flooding. In the emergency phase, it is a very useful tool to identify the areas subject to hydraulic risk and send relief. Here, we tested also its validity for the hydraulic model calibration. Based on the documentation provided on the web site, two images were used: Sentinel-1A/B (2017), acquired on 12 December 2017 at 07:58 UTC, and COSMO-SkyMed© ASI (2017), acquired on 14 December 2017 at 05:10 UTC, GSD 3 m of resolution, provided under COPERNICUS by the European Union and ESA. The resulting map in Figure 10a is an ensemble flood mapping elaborated through three independent state-of-the-art satellite flood mapping algorithms provided by the German Aerospace Center (DLR), Luxembourg Institute of Science and Technology (LIST) and Vienna University of Technology (TUW). The flood ensemble map is calculated at the pixel level and the classification of a pixel as flooded or not flooded is based on the majority (at least two out of three) of algorithms. By analyzing the Copernicus image, it is plausible to suppose that the first Sentinel-1 image corresponds to the image in Figure 3a, and it is used to map the inundation of the meanders in the upstream part of the river (in the South). Indeed, this SAR image does not show any flooded area after the levee breach. The COSMO-SkyMed image was likely used for simulating the big inundation due to the levee break (in the North of the image). If the Copernicus image is coupled with our model, the resulting $F$ index shows a maximum value for $n = 0.10$ m$^{-1/3}$s

(Figure 11), mainly due to the underestimation of the model in this southern part. The comparison map for this roughness coefficient value is illustrated in Figure 10b. Compared to the analysis carried out previously of the other images (Figure 7), the *F* index here is in between the one calculated by using the optical image and the Sentinel-1 image of 13 December. This condition is also due to the uncertainty in the selection of the right frame of the model to be compared with the Copernicus image, which is a composition of images.

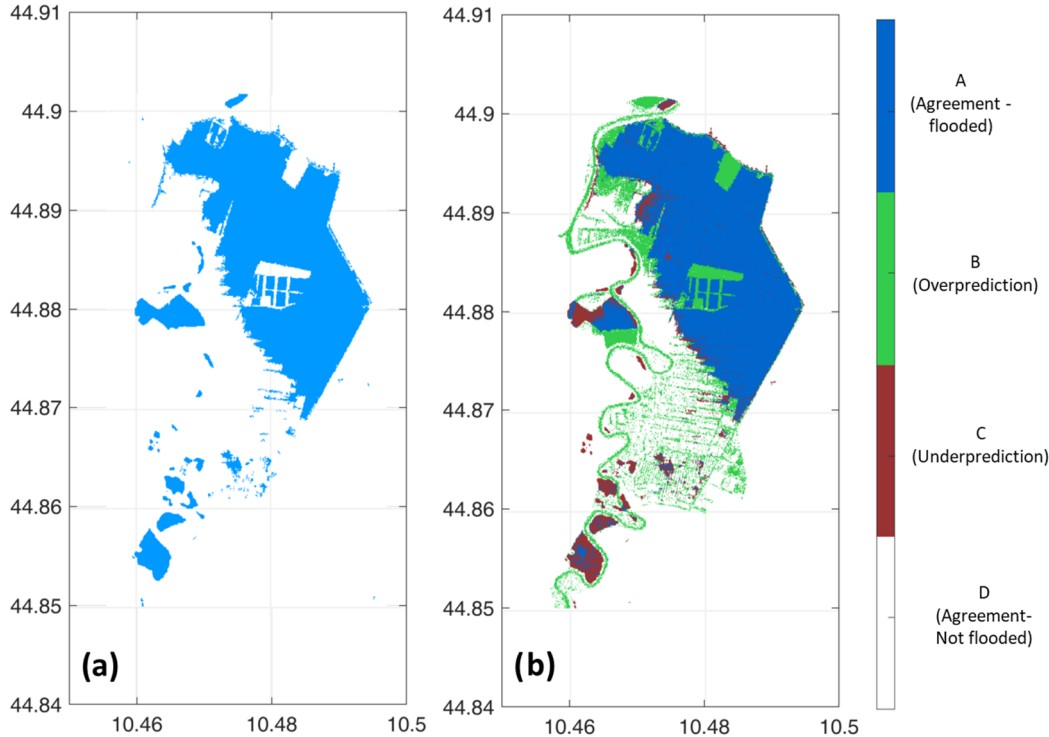

**Figure 10.** Flooded area extracted from the Copernicus Emergency Service for the event classified as EMSR260 in Cicognara, Sorbolo and Sant Ilario d'Enza [32] (**a**) and its comparison with the flood map derived from Wec-Flood model simulation with $n = 0.10 \ \text{m}^{-1/3}\text{s}$ (**b**).

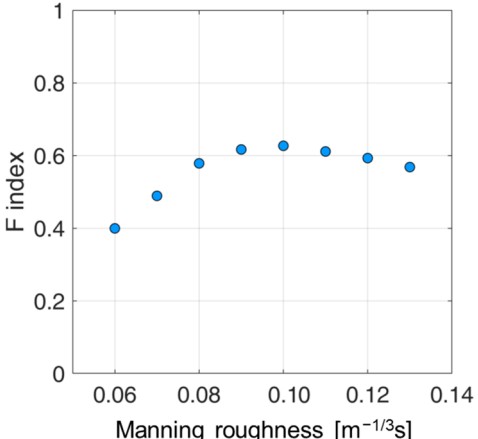

**Figure 11.** Variability of the measure of fit F as a function of the Manning roughness coefficient calculated with respect to the Copernicus Emergency Service flood map.

## 4. Discussion and Conclusions

As a demonstration case of a Digital Twin replica, we studied a levee failure that occurred along the Enza River causing flooding in the village of Lentigione di Brescello.

Several images were acquired during the flooding event, but only one SAR and one optical image were found suitable for the calibration of the hydraulic parameter (i.e., roughness coefficient). The results allowed us to conclude that the optical image, taken a few hours after the flood peak, proves to be more useful than the SAR image in assessing the limits of the extent of the flooded area, providing the most plausible results for calibrating the roughness value. However, this result cannot be considered valid in all cases. In fact, the main element in the evaluation of floods, such as the one studied in this work, is the timing of the satellite acquisition in relation to the event and the dynamics of the inundation. The images acquired during the recession limb could not optimally capture the maximum extent of the flooded area, as in this case study. Furthermore, the SAR image underestimated the flooded area mainly due to a visualisation problem with the modelled result. In fact, the hydraulic model returns the map as an envelope of the maximum extent of the flooded area and does not simulate the loss of water volume due to evaporation and infiltration that may have occurred on 13 December. The result is a flood map that is larger than the one actually observed by the satellite.

Therefore, the availability of satellite constellations capable of providing a more refined temporal coverage would be desirable because it would undoubtedly have a positive impact on the ability to monitor events' temporal evolution, hence, supporting the production of more accurate models.

The potential use of the Copernicus Emergency Map was also evaluated. Since the image comes from a multi-satellite (Sentinel-1 and COSMO-SkyMed) composition and the time of acquisition is not unique, this can cause difficulties in evaluating the frame to be extracted from the modelling maps. The source image for each pixel should be shown in order to understand the time of acquisition and the characteristics. However, even in this condition, the roughness coefficient can be calibrated in a plausible range, but it is necessary to have more information on the origin of the images.

The analyses carried out in this paper have shown the feasibility of creating a digital reality. The model developed can be adapted to investigate different conditions; if the levee breach is assumed to not occur, the model could provide flood scenarios caused by the overtopping process alone, or it could give information on the drought condition of the watercourse that has recently affected the Po basin. Using a rainfall-runoff model, the observed flow hydrograph at Sorbolo, used for the present study, can be replaced by modelled flows resulting from different combinations of precipitation, soil moisture and evaporation, allowing the construction of a "what if" scenario. As a result, several maps of flooded areas can be generated a priori and associated with a specific hydrograph to provide a comprehensive assessment of a specific flood event. This type of analysis is one of the main objectives of the Digital Twin, where the model is only a means to analyse different conditions that might occur in the future. Progress in this direction has been investigated under the DTE Hydrology project [39], where possible scenarios can be selected by the user and visualised in the form of hydrographs. A further step could be modelling the relative propagation of the flood and the associated flooded areas.

**Author Contributions:** Conceptualisation, S.B. and L.B.; methodology, A.T.; software, M.S., A.T.; validation, A.T., B.B. and S.B.; formal analysis, A.T., B.B. and S.B.; investigation, B.B. and A.T.; data curation, A.D.; writing—original draft preparation, A.T.; writing—review and editing, S.B., L.B., A.D., M.S.; visualisation, L.B.; supervision, S.B.; project administration, L.B.; funding acquisition, L.B. All authors have read and agreed to the published version of the manuscript.

**Funding:** This research was funded by DTE Hydrology Project, European Space Agency, grant number ESA 4000129870/20/I-NB-CCN N. 1.

**Data Availability Statement:** The hydraulic model is not freely available. Satellite image of Sentinel-1 and Sentinel-2 are available here (https://scihub.copernicus.eu/, last access on 20 April 2023), where the DTE demonstrator is available here (https://explorer.dte-hydro.adamplatform.eu/, accessed on 20 April 2023).

**Conflicts of Interest:** The authors declare no conflict of interest.

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
