# Peer review of "Flooding in the Digital Twin Earth: The Case Study of the Enza River Levee Breach in December 2017"

_water, doi:10.3390/w15091644_

Round 1

Reviewer 1 Report

Dear Editor and Authors,

the manuscript entitled "Flooding in the Digital Twin Earth: the case study of the Enza river levee breach in December 2017" is focused on the use of  Earth Observation to calibrate the roughness coefficients to be use in a 2D hydraulic modeling of the flooding event that involved a levee breach in December 2017 along the Enza River (Po River).

In my opinion this research fits the aim of the Journal and and can be valid to be published as the definition of the roughness coefficient, especially for mixed-bottom river beds, can be difficult to estimate.

However, the manuscript requires some improvements before publication in Water MDPI. Please refer to the attached file. 

Reviewer 2 Report

Please see my comments attached!

Reviewer 3 Report

The manuscript has an interesting topic that deals with a 2-D hydraulic model calibration procedure from a levee breaching event in December 2017, Italy using Earth Observation (EO) data. But, there are few question marks/drawbacks of the study as can be listed below:

1.       Several questions/comments/language/unclear parts have been marked on the text (see reviewer attachment)

2.       Especially on Figure 8 for the optical and Figure 9 for SAR image, the agreement/overprediction/underprediction colors seem to have many disagreements with the original images (Figure 5 and Figure 6). This could be due to: pixel resolution difference, projection mismatch or some other reasoning. This needs to be clarified and clearly explained. In doing so, adding extra figures showing only model flooded areas regarding the best Manning Coefficients may better answer this question.

3.       The cited references indicate that although there are twice as more usable SAR images compared to optical for inundated areas under flooding, the results of this study suggest that optical images provide better results. After this study, some firm conclusions have to be made clear whether the method (microwave/optical), algorithm, timing of the image or any other reasoning is behind the results.

4.       Figure 10 (flooded area extracted by the Copernicus Emergency Service) needs more elaboration. How this image is created, when it is prepared for usage, why it can’t/shouldn’t be utilized with hydraulic modeling? These questions need clear answers in order for the readers to better understand Copernicus Emergency Service products.

Round 2

Reviewer 2 Report

Good job!

Author Response

We woud like to thank the reviewer for the positive comment.

Reviewer 3 Report

The revised version of the manuscript has contributed significant changes to improve it. It is now a more comprehensible and flowing text with Figures. I still have a few minor comments (also indicated on the manuscript) that should be treated before publication as follows:

1.       The marked satellite lines on Figure 2. There seem to be 6 dates indicated where only 3 are used. Is there a certain reason for it to be given in this way?

2.       The inclusion of Copernicus flood mapping is treated positively, but the explanations on lines 401-403 seem a little confusing.

3.       Figure 10b is missing in the article

4.       Especially the Conclusion part of the text looks quite confusing with Track Changes view as some of the text is mistakenly included/excluded. The same goes for all the manuscript as well. Please carefully check which parts of the text in excluded from the old version and which are newly included.

5.       There are a few more minor language correction suggestions throughout the text.  
